# Factors Associated with the Early Initiation of Breastfeeding in Economic Community of West African States (ECOWAS)

**DOI:** 10.3390/nu11112765

**Published:** 2019-11-14

**Authors:** Osita Kingsley Ezeh, Felix Akpojene Ogbo, Garry John Stevens, Wadad Kathy Tannous, Osuagwu Levi Uchechukwu, Pramesh Raj Ghimire, Kingsley Emwinyore Agho

**Affiliations:** 1School of Science and Health, University of Western Sydney, Locked Bag 1797, Penrith, NSW 1797, Australia; Ezehosita@yahoo.com (O.K.E.); Prameshraj@hotmail.com (P.R.G.); 2Translational Health Research Institute (THRI), School of Medicine, Western Sydney University, Campbelltown Campus, Locked Bag 1797, Penrith, NSW 2571, Australia; F.Ogbo@westernsydney.edu.au; 3Humanitarian and Development Research Initiative (HADRI), School of Social Sciences and Psychology, Western Sydney University, Locked Bag1797, Penrith, NSW 2751, Australia; G.Stevens@westernsydney.edu.au; 4School of Business, Western Sydney University, Locked Bag 1797, Penrith, NSW 2751, Australia; K.Tannous@westernsydney.edu.au; 5Diabetes, Obesity and Metabolism Translational Research Unit, Western Sydney University, Campbelltown, NSW 2560, Australia; L.osuagwu@westernsydney.edu.au

**Keywords:** breastfeeding, infants, ECOWAS, antenatal care, pregnancy, infant mortality

## Abstract

The early initiation of breastfeeding (EIBF) within one hour after birth enhanced mother–newborn bonding and protection against infectious diseases. This paper aimed to examine factors associated with EIBF in 13 Economic Community of West African States (ECOWAS). A weighted sample of 76,934 children aged 0–23 months from the recent Demographic and Health Survey dataset in the ECOWAS for the period 2010 to 2018 was pooled. Survey logistic regression analyses, adjusting for country-specific cluster and population-level weights, were used to determine the factors associated with EIBF. The overall combined rate of EIBF in ECOWAS was 43%. After adjusting for potential confounding factors, EIBF was significantly lower in Burkina Faso, Cote d’Ivoire, Guinea, Niger, Nigeria, and Senegal. Mothers who perceived their babies to be average and large at birth were significantly more likely to initiate breastfeeding within one hour of birth than those mothers who perceived their babies to be small at birth. Mothers who had a caesarean delivery (AOR = 0.28, 95%CI = 0.22–0.36), who did not attend antenatal visits (ANC) during pregnancy, and delivered by non-health professionals were more likely to delay initiation of breastfeeding beyond one hour after birth. Male children and mothers from poorer households were more likely to delay introduction of breastfeeding. Infant and young child feeding nutrition programs aimed at improving EIBF in ECOWAS need to target mothers who underutilize healthcare services, especially mothers from lower socioeconomic groups.

## 1. Introduction

Globally, about 75% of all under-five deaths occur within the first year of life, and sub-Saharan African countries remain the hub of the world burden of all under-five deaths [1,2]. Early initiation of breastfeeding (EIBF) within one hour of birth remains a significant public health challenge, particularly in sub-Saharan African (SSA) countries, including the Economic Community of West African States (ECOWAS). In keeping with the Sustainable Development Goals, developing countries have committed to reducing under-five mortality by 25 deaths per 1000 live births by 2030 [3]. Global strategies, including the Baby-Friendly Hospital Initiative (BFHI), Community Integrated Management of Childhood Illness (C-IMCI), and Infant and Young Child Feeding (IYCF) guidelines have been developed by United Nations Children’s Fund (UNICEF) and the World Health Organization (WHO). These aim to substantially reduce child mortality, morbidity, and undernutrition [4,5]. They have succeeded in some parts of the world but remain refractory in Africa. In addition, there is a strong epidemiological evidence to suggest that the nutritional benefits of EIBF may include a reduced risk of hypoglycemia in at-risk infants (e.g., small for gestational age and macrosomia infants), which can cause significant morbidity and mortality [6]. This is due to EIBF containing antibodies for natural immunity, which provide protection against infectious diseases (such as diarrheal and pneumonia) and other potentially life-threatening ailments [7,8]. ECOWAS is a regional political and economic union of 15 countries in West Africa, founded in 1975 with members including Benin, Burkina Faso, Cabo Verde, Cote d’Ivoire, The Gambia, Ghana, Guinea, Guinea-Bissau, Liberia, Mali, Niger, Nigeria, Senegal, Sierra Leone, and Togo. The main aim of this alliance is to promote socioeconomic integration among member states to raise living standards and promote economic development, with wider implications for judicial and health service cooperation [9].

There are numerous recognized advantages of EIBF, however, the rates of EIBF in ECOWAS are typically low and still suboptimal, ranging from 17% to 62% [10,11]. These low rates mean a substantial proportion of newborn babies are deprived of mother–newborn bonding, resistance against infection, and the early breastfeeding needed to fight disease [7,8]. Studies highlight that infants initiated to breastfeeding within an hour of birth have a reduced incidence of life-threatening conditions, such as serious systemic bacterial infections, diarrhea, respiratory infections, as well as less severe infections of the middle ear and urinary tract, and allergic conditions [12,13,14]. A cohort study conducted in rural Ghana indicated that 22% of neonatal deaths could be prevented if breastfeeding started within the first hour of birth [15]. Similarly, a community-based randomized trial conducted in southern Nepal revealed that approximately 8–19% of newborn deaths could be averted annually in developing countries if all newborns were breastfed within an hour of delivery [16]. EIBF is not only beneficial to neonates but also to mothers, as it reduces the risk of postpartum hemorrhage [17]. 

Published studies conducted on EIBF in SSA, including ECOWAS countries, have indicated that EIBF is associated with demographic, socioeconomic, and obstetric characteristics. A nationwide cross-sectional study on trends and factors associated with EIBF in Nigeria found that mothers who had four or more antenatal visits (ANC) visits during pregnancy, mothers from wealthier households, and mothers who delivered their babies at the health facility are more likely to initiate breastfeeding within one hour of birth [18,19]. Another population-based study conducted in Nigeria that examined urban–rural differences in the rates of EIBF concluded that EIBF rates were significantly lower in rural areas than urban areas [20]. The perception of a lack of breastmilk, perception that the mother and the baby need rest after birth, performing post-birth activities (such as bathing), and the baby not crying for milk were observed as barriers to EIBF among mothers in Ghana [11]. In that study, delivery in a health facility was a facilitator to EIBF, as health practitioners encouraged early breastfeeding [11]. A nationwide cross-sectional study in Zimbabwe found that skilled delivery assistants, multiparity, and postpartum skin to skin contact between mother and newborn babies immediately after birth were significantly related to EIBF [21]. Similarly, in a community-based cross-sectional study on EIBF conducted in Ethiopia, maternal education, delivery at a health facility and visiting antenatal care services during pregnancy were also significantly associated with EIBF within one hour of delivery [22]. The low rate of EIBF practice in ECOWAS could have potentially contributed to nearly half of all under-five deaths, such as those due to prematurity or malnutrition in sub-Saharan Africa region in the year 2017 [1]. Prior to the current study, no studies have used combined country-specific local data across ECOWAS countries to examine factors associated with EIBF within one hour of birth. Evidence based on collective data will provide region-specific targeted interventions to scale up EIBF practices to reduce morbidity and mortality. 

Hence, this study aims to investigate the possible characteristics influencing EIBF in 13 of the ECOWAS member countries using the most recent Demographic and Health Survey (DHS) survey datasets conducted between 2010 and 2018. Recognition of factors associated with EIBF in ECOWAS countries is essential for building effective nutrition education and behavior change communication interventions in targeting women at heightened risk of sub-optimal feeding behaviors. In addition, the identification of modifiable associated factors will help public health nutrition program managers and policymakers develop strategies to support these contributory factors. ECOWAS would gain enormous health and economic benefits by improving EIBF. Thus, the findings from this study could assist in the planning and monitoring of effective infant and breastfeeding promotion programs in each ECOWAS country, which may contribute to achieving the child survival sustainable development goal.

## 2. Materials and Methods 

The most recent ECOWAS DHS datasets between 2010 and 2018 were used for this study. Cabo Verde and Guinea-Bissau were not included in the study because their last DHS surveys were not within the study period. As a result, the most recent DHS datasets from 13 ECOWAS countries were used for this study. These data sets were obtained from standardized population-based cross-sectional surveys with a high response rate and trained interviewers [10]. The DHS surveys are publicly available for download [10,23]. Each ECOWAS National Bureau of Statistics collects the DHS Data in collaboration with the United States Agency for International Development [10]. The DHS surveys use identical questionnaires in all countries to collect information on diverse topics, including fertility, reproductive health, maternal and child health, mortality, nutrition, and self-reported health behaviors among adults [5]. In this study and to reduce recall bias, we restricted the analysis to the last born child at 23 months and living with the respondent. This yielded a weighted total of 76,934 children.

### 2.1. Outcome and Confounding Factors

The outcome variable was EIBF. Women were asked how long after birth the baby was put to the breast for the first time. Responses were recorded in minutes and/or hours. EIBF within one hour of birth was estimated using the World Health Organization recommendation [24,25]. This indicator includes the EIBF rate (the proportion of children born aged 0–23 months who were put to the breast within one hour of birth). EIBF was categorized into a binary form of the outcome variables: “Yes” (1 = if a child was given EIBF) and “No” (0 = if a child was not given EIBF). The potential confounding factors were organised into four distinct groups: demographic factors (countries, place of residence, mother’s age, marital status, combined birth rank and birth interval, sex of baby, child’s age in category, perceived size of the baby); socio-economic factors (household wealth index, maternal work in the last 12 months, maternal education, and maternal literacy); access to media factors (frequency of reading newspapers/magazine, frequency of listening to radio, and frequency of watching television); and healthcare utilization factors (place of delivery, mode of delivery, type of delivery assistance, and antenatal clinic visits). For the pooled dataset, the household wealth index was constructed using the ‘hv271’ variable. The hv271 is a household’s wealth index value generated by the product of standardized scores (z-scores) and factor coefficient scores (factor loadings) of wealth indicators [26]. In the household wealth index categories, the bottom 20% of households was arbitrarily referred to as the poorest households and the top 20% as richest households, and they were divided into poorest, poor, middle, rich, and richest. 

### 2.2. Statistical Analysis

For the combined 13 ECOWAS datasets, a population-level weight, unique country-specific clustering, and strata were created. This was to manage large country weights being greater than small country weights and also because clusters were different in each country. Population-level weights were used for survey (SVY) tabulation that adjusted for a unique country-specific stratum, and clustering was used to determine the percentage, frequency count, and univariate analysis of all selected characteristics. Country-specific weights were used for the Taylor series linearization method [27] in the surveys when estimating the rates and 95% confidence intervals of EIBF in each country.

Population-level weights were used in the multivariate analyses. In the multivariate analyses, a four-stage hierarchical model was carried out. In the first stage model, demographic factors were entered into the model. In the second stage model, demographic factors were added to the socio-economic factors. A similar procedure was employed for the third stage model, which included the demographic and socio-economic level factors, as well as access to media factors. The fourth and final stage model, healthcare utilization factors, were added to demographic, socio-economic, and media factors. All statistical analyses were conducted using (statistics and data) STATA/MP Version.14.1 (StataCorp, College Station, TX, USA) and adjusted odds ratios (AORs) and their 95% confidence intervals (CIs) obtained from the adjusted multivariate logistic regression model were used to measure the factors associated with EIBF.

## 3. Results

### 3.1. Characteristics of the Sample and Univariate Analyses for EIBF

The pooled rate of EIBF in the 13 ECOWAS was 43% and EIBF lay between 52% and 62% in Togo, Niger, Mali, Benin, Sierra Leone, Liberia, Ghana, and Gambia, as indicated in Figure 1. The highest rates of EIBF were observed in Togo and Liberia, and the lowest was in Guinea (17%). 

Table 1 illustrates the characteristics of the sample of children aged 0–23 months and the unadjusted odd ratios of EIBF in children aged 0–23 months. Overall, 21% of children aged 0–23 months were from Nigeria, and compared to Benin (7.7%), the odds of EIBF in Nigeria was reduced significantly, by 58% (OR = 0.42, 95% CI:0.36, 0.48). About 42% of EIBF babies were perceived by their mothers to be of average birth size compared to the 18% of EIBF babies perceived to be small at birth. The unadjusted odd ratios of EIBF were significant with babies perceived by their mothers to be the average birth size at birth, combined birth rank and birth interval, household wealth, frequency of listening to the radio, and watching television. 

### 3.2. Factors Associated with Early Initiation of Breastfeeding

As shown in Table 2, EIBF within one hour of birth was significantly lower among children living in Nigeria, Cote d’Ivoire, Burkina Faso, Guinea, and Senegal. Mothers from poorest households, mothers’ first birth, and delivery by non-health professionals were significantly associated with delayed initiation of breastfeeding. Mothers who underwent caesarean delivery were significantly more likely to delay the initiation of breastfeeding (adjusted OR = 0.28, 95% CI:0.22, 0.36; *p* < 0.001) than those who had non-caesarean delivery.

Mothers who perceived their babies to be average or larger at birth were more likely to initiate breastfeeding within one hour of delivery than mothers who perceived their babies to be small. EIBF was significantly higher among mothers who frequently listened to the radio, who delivered their babies at the health facilities, and those mothers who had four or more antenatal clinic visits. EIBF was significantly higher with female births (adjusted OR = 1.17, 95% CI:1.00, 1.37; *p* = 0.046) than male births.

## 4. Discussion 

Our results indicate that EIBF rates in the 13 ECOWAS are at a sub-optimum level, and need further improvement to fulfil the goal of optimal infant feeding in this subregion and the related Sustainable Development Goals (SDGs) of reducing neonatal mortality. During the study period (2010–2018), we noted the uneven prevalence of EIBF among infants aged between 0 and 23 months of age across the ECOWAS countries, ranging from a low of 17% (Guinea) to the highest of 61% (Togo). The reported rates of EIBF in this study are still well below the expected target of ≥70% recommended by the WHO [1] to further reduce infant mortality and morbidity in the sub-region by 25 deaths per 1000 live births by 2030. The wide variation in the rates of EIBF within the 13 ECOWAS countries may be due to socio-cultural, geographical location, and health inequalities, as well as economic issues among different populations. The study indicated that place of delivery (health institution), mode of delivery (vaginal delivery), ANC visits (≥4), and household wealth index (poorer or middle-class household) were significantly associated with EIBF within one hour of birth. Media exposure (newspaper/magazine or radio), baby size at birth (average and large), child’s sex (female), and first birth order were also found to be positively related to EIBF. Delayed introduction to breastfeeding was significant in six ECOWAS member states (Gambia, Nigeria, Senegal, Guinea, Burkina Faso, and Cote d’Ivoire).

Child delivery at health facilities predicts EIBF practices within one hour of the birth. This finding is consistent with previous studies conducted in Nepal [28], Nigeria [29], Namibia [30], and Ethiopia [31]. However, this is not unexpected given that the majority of health institutions across ECOWAS countries have adopted BFHI, which is overseen by skilled health providers (midwife, nurse, or doctor) whose core aim is to encourage and assist mothers in achieving optimal breastfeeding practices, including EIBF. It has been suggested that the presence of trained breastfeeding and delivery assistants in health institutions improved mothers’ EIBF practice [32]. This may have contributed to the substantial proportion (60%) of EIBF among mothers who gave birth at a hospital, as observed in the current study.

Similar to reports from earlier studies [22,29,30], the present study found that vaginal delivery was strongly correlated with EIBF practice when compared with caesarean birth. A plausible reason for this difference may be related to the adverse physical effects mothers and newborns often experience after CS surgery. Factors such as excruciating pain, anesthesia, prolonged labor, and related respiratory distress among newborns [33,34] may result in healthcare providers being preoccupied with helping both babies and mothers to stabilize rather than initiating breastfeeding [35,36]. 

Previously published studies indicated that pregnant women being unable to access or attend ANC services during pregnancy poses a significant barrier in initiating breastfeeding within one-hour post-birth [37,38]. However, in the present study, we found that mothers who had four or more ANC visits to a health institution prior to childbirth showed a greater likelihood of EIBF practice than those who had none or less than four ANC visits. This result is similar to that reported in past studies carried out in India [39] and Nepal [40]. Breastfeeding counseling and health messages received through the Baby-Friendly Hospital Initiative (BFHI) may have contributed to the increased likelihood of EIBF noted in the study. It has been suggested in previous work that breastfeeding counseling during ANC visits is positively related to the mother’s adherence to the WHO optimal breastfeeding practices [41,42]. Therefore, promoting and supporting ANC uptake in ECOWAS countries could ensure remarkable improvement in EIBF practices. 

Newborn babies whose mother perceived their size to be average or larger had an increased likelihood of EIBF within one hour of birth compared with those who were perceived to be small or smaller size. This finding is consistent with a cross-sectional study carried out in Nigeria in 2016, which indicated that newborn babies perceived by their mothers as average or larger size at birth were more likely to be EIBF [29]. Also, a cross-sectional study done in Nepal in 2014 reported that newborn babies whose size was large at birth were significantly more likely to receive EIBF within one hour of birth in comparison with babies whose size was small [28]. The significantly increased odds of EIBF practice among newborn babies perceived to be of average or larger size in ECOWAS countries may be attributed to their physical maturity and associated proper coordination of the suction–deglutination–respiratory cycle and breast-seeking reflex [28]. Conversely, the immaturity of small babies often hinders the abilities required to initiate breastfeeding within one hour of delivery due to poor breast-seeking reflex and sucking inability [43]. Additionally, illness often leads to newborn babies being kept away from their mothers [44], resulting in the early introduction of complementary drinks. Notably, female births were found to be associated with EIBF within one hour of delivery. This outcome is similar to earlier reports in Sri Lanka [45] and Ethiopia [46], which found that male babies were less likely to receive early breastfeeding than their female counterparts. This gender difference in EIBF practice may be influenced by socio-cultural beliefs and require further assessment. However, previous studies have suggested that the practice of prelacteal feeds (or the act of given foods to newborn babies before initiating breastfeeding) in male babies is common and acceptable in African [8,47] and Asian countries [47,48]. 

Newborn babies born to mothers from a poor household had significantly higher odds of initiating breastfeeding within one hour of birth compared with those from a wealthy household. This finding contradicts previous reports, which indicated that mothers from wealthy households were more likely to initiate breastfeeding in the first hour of delivery [28,35]. Nevertheless, our finding is similar to those reported from Nepal [49] and Namibia [30]. The significantly higher odds of EIBF practice among poor mothers noted in the current study may be attributed to lack of access to infant formula or inability to buy costly complementary substitutes to breast milk, which may result in poor mothers depending solely on breastfeeding. It has also been suggested that mothers from wealthy households were more likely have an elective caesarean section, resulting in delayed initiation of breastfeeding within one hour of birth [50]. It was not unexpected that mothers who frequently watch television or listen to radio were more likely to initiate breastfeeding within one hour of delivery. This finding is consistent with previous studies [51,52]. Apart from health facilities, exposure to other information sources regarding breastfeeding practices during antenatal and postnatal care can also play a key role in encouraging women to practice early initiation of breastfeeding.

Similar to previous reports [30,46], we found that mothers who were having their first child were less likely to initiate breastfeeding within one hour of birth compared with those with two or more deliveries. This may be related to inexperience regarding breastfeeding and relatively poor use of maternal health services by first time mothers. As shown in a past study, earlier breastfeeding experience is correlated with both intention and timely breastfeeding initiation [53]. 

## 5. Limitations and Strengths

The following limitations of this study should be considered when making specific conclusions about its findings: (a) the study was unable to establish a causal relationship between the study characteristics and EIBF, as cross-sectional data was used to identify the predictors of EIBF; (b) recall and measurement errors may have overestimated or underestimated the findings of this study, as data regarding some of the study factors was obtained from mothers up to five years after childbirth; (c) there were some unmeasured coexisting factors, such as instrumental vaginal birth, cultural beliefs, and health professionals’ prior knowledge of EIBF or family dynamics, which may have also impacted the study findings. Additionally, detailed information concerning infant formula advertising via electronic or print media was not available at the time of the DHS survey. Despite the highlighted weaknesses, there are a number of strengths that should also be noted regarding this study. Firstly, the rate of EIBF was based on nationally representative data of 13 countries, and hence findings could be generalized across the subregion. Secondly, the results are comparable across the ECOWAS countries because all the variables used were similarly described across countries. Thirdly, the study used a combined nationally representative large sample size across the 13 ECOWAS, which meant that the potential impact of selection bias was minimal and unlikely to affect the estimates. 

## 6. Conclusions

The findings of this study suggest that delivery in a hospital or health facility, vaginal delivery, frequent ANC visits (≥4), middle income household, access to electronic media (television or radio), average or larger baby size at birth, child’s sex (female), and first birth order were significantly associated with EIBF within one hour of birth in the 13 ECOWAS countries. In the ECOWAS subregion, it is crucial that integrated nutritional health promotion campaigns be conducted both at the community and individual levels. At the community level, the role of midwives and community health officers (CHO) in promoting breastfeeding practices appeared evident, with CHO antenatal home visits contributing to the early initiation of breastfeeding and empowering the community with greater knowledge of the nutrition and health requirements of children under two years of age. At the individual level, interventions to promote the involvement of family members during pregnancy and child birth are needed in ECOWAS and should particularly target young mothers and families from lower socioeconomic groups with lower rates of EIBF.

## Figures and Tables

**Figure 1 nutrients-11-02765-f001:**
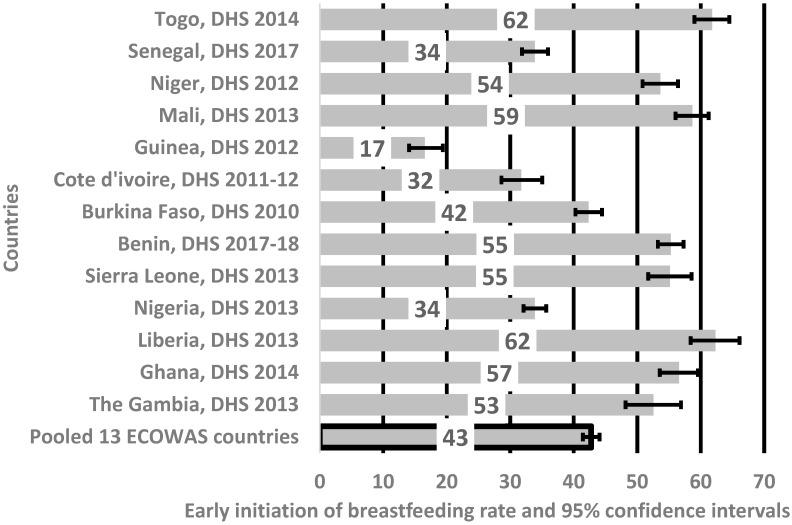
Rates of early initiation of breastfeeding in thirteen Economic Community of West African States. DHS = Demographic and Health Survey.

**Table 1 nutrients-11-02765-t001:** Individual, household, and community level characteristics and unadjusted odds ratios (OR) (95% CI) of early initiation of breastfeeding (EIBF) among children aged 0–23 months in 13 Economic Community of West African States (ECOWAS).

Characteristic	*n* *	% *	%	EIBF			
				OR	95% CI		*p*-Value
*Demographic factors*							
Country							
Benin	5937	7.7	9.2	1.00			
Burkina Faso	6887	9.0	10.0	0.58	0.50	0.67	<0.001
Cote d’ivoire	4554	5.9	5.2	0.43	0.33	0.56	<0.001
The Gambia	5387	7.0	6.0	0.76	0.52	1.11	0.151
Ghana	3412	4.4	4.0	1.07	0.88	1.31	0.483
Guinea	3574	4.6	4.7	0.17	0.13	0.22	<0.001
Liberia	4241	5.5	5.1	1.30	0.96	1.77	0.093
Mail	4843	6.3	6.7	1.11	0.93	1.32	0.242
Niger	7044	9.2	8.1	0.90	0.75	1.08	0.262
Nigeria	15,993	20.8	20.7	0.42	0.36	0.48	<0.001
Senegal	5790	7.5	8.1	0.35	0.29	0.41	<0.001
Sierra Leone	6230	8.1	7.4	0.85	0.67	1.08	0.173
Togo	3042	4.0	4.8	1.49	1.23	1.79	<0.001
Residence							
Urban	27,769	36.1	31.0	1.00			
Rural	49,165	63.9	69.0	0.96	0.85	1.08	0.468
Mothers’ age in years							
15–19	25,575	33.2	33.4	1.00			
20–34	36,855	47.9	46.9	1.09	1.02	1.17	0.012
35–49	14,503	18.9	19.8	1.08	0.99	1.17	0.084
Marital status							
Currently married	70,916	92.2	92.7	1.00			
Formerly married ^	1875	2.4	2.5	1.13	0.91	1.39	0.269
Never married	4142	5.4	4.8	0.98	0.83	1.17	0.863
Age of child (months)							
0–5	19,736	25.7	25.8	1.00			
6–11	20,308	26.4	26.9	1.02	0.93	1.12	0.657
12–17	20,402	26.5	26.0	1.07	0.98	1.16	0.138
18–23	16,488	21.4	21.3	1.10	1.00	1.21	0.047
Birth order or birth rank							
First-born	15,935	20.7	20.0	1.00			
Second–fourth	36,840	47.9	46.6	1.26	1.16	1.36	<0.001
Fifth or more	24,159	31.4	33.4	1.19	1.09	1.31	<0.001
Preceding birth interval (*n* = 76,748)							
No previous birth	15,935	20.7	20.0	1.00			
<24 months	8163	10.6	10.6	1.13	1.00	1.28	0.042
≥24 months	52,650	68.4	69.2	1.25	1.16	1.35	<0.001
Combined birth rank and birth interval							
Second/third birth rank, more than two years interval	30,717	39.9	39.0	1.00			
First birth rank	15,935	20.7	20.0	0.78	0.72	0.85	<0.001
Second/third birth rank, less than or equal to two years interval	6123	8.0	7.7	0.90	0.79	1.02	0.097
Fourth birth rank, more than two years interval	20,520	26.7	28.4	0.94	0.87	1.02	0.120
Fourth birth rank, less than or equal to two years interval	3638	4.7	5.0	0.89	0.77	1.02	0.103
Sex of baby							
Male	38,723	50.3	50.6	1.00			
Female	38,211	49.7	49.4	1.05	0.99	1.11	0.120
Size of baby (*n* = 76,255)							
Small	13,689	17.8	17.5	1.00			
Average	32,517	42.3	42.6	1.22	1.11	1.34	<0.001
Large	30,049	39.1	38.9	1.19	1.07	1.31	0.001
*Socio-economic factors*							
Household Wealth Index							
Poorest	12,753	16.6	20.1	1.00			
Poorer	13,520	17.6	19.7	1.35	1.19	1.54	<0.001
Middle	15,088	19.6	19.8	1.55	1.36	1.78	<0.001
Richer	14,854	19.3	20.1	1.33	1.16	1.52	<0.001
Richest	20,719	26.9	20.5	1.25	1.07	1.46	0.005
Work in the last 12 months (*n* = 76,918)							
Not working	30,637	39.8	38.2	1.00			
Working	46,281	60.2	61.8	0.94	0.86	1.02	0.150
Maternal education (*n* = 76,926)							
No education	45,527	59.2	61.6	1.00			
Primary	13,319	17.3	18.1	1.06	0.96	1.17	0.239
Secondary and above	18,080	23.5	20.4	1.03	0.93	1.14	0.549
Maternal Literacy (*n* = 76,549)							
Cannot read at all	57,292	74.5	77.4	1.00			
Able to read only part of sentences	19,257	25.0	22.1	0.96	0.88	1.04	0.341
*Access to media*							
Frequency of reading newspaper or magazine (*n* = 76,736)							
Not at all	68,490	89.0	91.5	1.00			
Less than once a week	4575	5.9	4.8	0.93	0.81	1.08	0.356
At least once a week	3644	4.7	3.7	1.10	0.92	1.32	0.281
Almost every day	27	0.0	0.0	2.13	0.80	5.69	0.132
Frequency of listening to radio (*n* = 76,808)							
Not at all	25,977	33.8	36.0	1.00			
Less than once a week	19,798	25.7	25.0	0.95	0.87	1.04	0.300
At least once a week	30,250	39.3	37.8	0.97	0.88	1.06	0.466
Almost every day	783	1.0	1.1	3.27	2.49	4.29	<0.001
Frequency of watching Television (*n* = 76,760)							
Not at all	42,468	55.2	59.8	1.00			
Less than once a week	12,287	16.0	15.1	1.01	0.90	1.13	0.861
At least once a week	21,294	27.7	24.1	0.84	0.76	0.93	0.001
Almost every day	711	0.9	0.8	2.60	1.97	3.44	<0.001
*Healthcare utilization factors*							
Place of delivery							
Home	31,183	40.5	41.3	1.00			
Health facility	45,751	59.5	58.7	1.34	1.23	1.47	<0.001
Mode of delivery (*n* = 76,729)							
Non-caesarean	73,763	95.9	96.3	1.00			
Caesarean section ^ⱡ^	2966	3.9	3.5	0.39	0.32	0.49	<0.001
Type of delivery assistance (*n* = 63,553)							
Health professional	49,991	65.0	62.8	1.00			
Traditional birth attendant	1884	2.4	2.8	0.73	0.61	0.88	0.001
Other untrained	6959	9.0	10.7	0.76	0.64	0.90	0.001
No one	4699	6.1	6.0	0.61	0.51	0.72	<0.001
Antenatal clinic visits (*n* = 74,550)							
None	9478	12.3	12.6	1.00			
1–3	23,314	30.3	31.9	1.55	1.36	1.77	<0.001
4+	41,758	54.3	52.7	1.67	1.45	1.91	<0.001

^ divorce/separated/widowed; * Weighted sample sizes and percentages; ^ⱡ^ Caesarean section is a combination of both general and regional anesthesia.

**Table 2 nutrients-11-02765-t002:** Adjusted odds ratio (AOR) (95% confidence intervals (CI)) of factors associated with the early initiation of breastfeeding (EIBF) among children aged 0–23 months in 13 Economic Community of West African States (ECOWAS).

Characteristic	Model 1				Model 2				Model 3				Model 4			
	AOR	95% CI		*p*-Value	AOR	95% CI		*p*-Value	AOR	95% CI		*p*-Value	AOR	95% CI		*p*-Value
*Demographic factors*																
Benin	1.00				1.00				1.00				1.00			
Burkina Faso	0.58	0.50	0.67	<0.001	0.56	0.48	0.65	<0.001	0.56	0.48	0.66	<0.001	0.58	0.48	0.69	<0.001
Cote d’ivoire	0.42	0.32	0.54	<0.001	0.43	0.33	0.56	<0.001	0.43	0.33	0.56	<0.001	0.43	0.33	0.55	<0.001
The Gambia	0.75	0.51	1.09	0.134	0.73	0.50	1.07	0.109	0.75	0.51	1.11	0.148	0.67	0.44	1.02	0.062
Ghana	1.08	0.88	1.34	0.460	1.11	0.89	1.38	0.351	1.15	0.92	1.43	0.212	1.33	1.03	1.72	0.028
Guinea	0.17	0.13	0.22	<0.001	0.17	0.13	0.23	<0.001	0.17	0.13	0.23	<0.001	0.19	0.14	0.25	<0.001
Liberia	1.30	0.95	1.79	0.106	1.34	0.96	1.87	0.081	1.34	0.96	1.87	0.085	1.44	1.04	1.98	0.028
Mali	1.10	0.93	1.32	0.271	1.30	1.07	1.58	0.009	1.34	1.09	1.63	0.005	1.36	1.09	1.71	0.006
Niger	0.94	0.78	1.13	0.520	0.86	0.71	1.04	0.132	0.87	0.72	1.05	0.144	0.98	0.80	1.22	0.882
Nigeria	0.41	0.36	0.48	<0.001	0.44	0.38	0.51	<0.001	0.44	0.38	0.52	<0.001	0.51	0.43	0.61	<0.001
Senegal	0.36	0.30	0.43	<0.001	0.37	0.31	0.45	<0.001	0.38	0.32	0.46	<0.001	0.40	0.33	0.48	<0.001
Sierra Leone	0.86	0.67	1.10	0.230	0.84	0.65	1.07	0.151	0.83	0.65	1.07	0.147	0.78	0.60	1.03	0.075
Togo	1.50	1.25	1.81	0.000	1.50	1.24	1.81	<0.001	1.29	1.05	1.59	0.015	1.56	1.25	1.94	<0.001
Residence																
Urban	1.00				1.00				1.00				1.00			
Rural	0.95	0.84	1.06	0.335	0.94	0.83	1.07	0.373	0.94	0.82	1.07	0.355	0.92	0.81	1.04	0.200
Mothers’ age in years																
15–19	1.00				1.00				1.00				1.00			
20–34	0.98	0.90	1.07	0.732	0.99	0.91	1.08	0.777	0.98	0.90	1.07	0.605	0.98	0.90	1.08	0.731
35–49	0.98	0.86	1.11	0.725	0.98	0.86	1.11	0.727	0.96	0.85	1.09	0.538	0.94	0.82	1.08	0.372
Marital status																
Currently married	1.00				1.00				1.00				1.00			
Formerly married ^	0.99	0.79	1.26	0.957	0.99	0.78	1.26	0.940	0.99	0.78	1.26	0.952	0.98	0.76	1.26	0.894
Never married	0.96	0.78	1.18	0.692	0.93	0.76	1.14	0.501	0.95	0.77	1.17	0.610	0.98	0.79	1.21	0.843
Birth rank and birth interval																
Second/third birth rank, more than two years interval	1.00				1.00				1.00				1.00			
First birth rank	0.78	0.71	0.86	<0.001	0.78	0.70	0.86	<0.001	0.77	0.70	0.85	<0.001	0.78	0.69	0.87	<0.001
Second/third birth rank, less than or equal to two years interval	0.89	0.78	1.03	0.123	0.89	0.77	1.02	0.098	0.89	0.77	1.03	0.105	0.91	0.78	1.07	0.268
Fourth birth rank, more than two years interval	0.98	0.89	1.08	0.699	0.99	0.89	1.09	0.791	1.00	0.90	1.11	0.978	1.03	0.93	1.15	0.551
Fourth birth rank, less than or equal to two years interval	0.92	0.79	1.07	0.268	0.92	0.78	1.07	0.288	0.93	0.79	1.09	0.359	0.98	0.82	1.17	0.808
Sex of baby																
Male	1.00				1.00				1.00				1.00			
Female	1.07	1.01	1.13	0.027	1.07	1.01	1.13	0.024	1.07	1.01	1.13	0.025	1.08	1.01	1.16	0.021
Age of child (months)																
0–5	1.00				1.00				1.00				1.00			
6–11	1.03	0.94	1.13	0.546	1.03	0.94	1.13	0.538	1.03	0.93	1.13	0.570	1.02	0.92	1.14	0.664
12–17	1.09	1.00	1.20	0.056	1.10	1.00	1.20	0.045	1.09	0.99	1.19	0.071	1.10	0.98	1.22	0.096
18–23	1.10	1.00	1.21	0.048	1.10	1.00	1.21	0.047	1.10	1.00	1.21	0.052	1.10	0.99	1.23	0.075
Size of baby																
Small	1.00				1.00				1.00				1.00			
Average	1.21	1.09	1.33	<0.001	1.20	1.08	1.32	<0.001	1.20	1.09	1.32	<0.001	1.19	1.07	1.32	0.002
Large	1.23	1.11	1.37	<0.001	1.22	1.10	1.36	<0.001	1.22	1.10	1.36	<0.001	1.20	1.07	1.36	0.002
*Socio-economic factors*																
Household Wealth Index																
Poorest					1.00				1.00				1.00			
Poorer					1.29	1.13	1.48	<0.001	1.32	1.15	1.51	<0.001	1.19	1.02	1.38	0.028
Middle					1.38	1.20	1.59	<0.001	1.42	1.23	1.63	<0.001	1.22	1.03	1.43	0.018
Richer					1.34	1.17	1.55	<0.001	1.39	1.20	1.60	<0.001	1.17	0.98	1.39	0.077
Richest					1.16	0.98	1.36	0.083	1.19	1.00	1.43	0.052	1.03	0.84	1.26	0.782
Work in the last 12 months																
Not working					1.00				1.00				1.00			
Working					0.95	0.86	1.04	0.243	0.95	0.87	1.04	0.291	0.95	0.87	1.05	0.329
Maternal education																
No education					1.00				1.00				1.00			
Primary					1.02	0.92	1.13	0.732	1.03	0.92	1.14	0.627	1.00	0.90	1.12	0.992
Secondary and above					0.98	0.80	1.21	0.860	0.96	0.78	1.19	0.717	0.89	0.73	1.10	0.291
Maternal Literacy																
Cannot read at all					1.00				1.00				1.00			
Able to read only part of sentences					1.04	0.87	1.24	0.690	1.00	0.82	1.21	0.985	1.02	0.85	1.23	0.811
*Access to media*																
Frequency of reading newspaper or magazine																
Not at all									1.00				1.00			
Less than once a week									1.11	0.93	1.32	0.255	1.06	0.89	1.27	0.508
At least once a week									1.33	1.09	1.63	0.006	1.36	1.10	1.67	0.004
Almost every day									0.91	0.33	2.51	0.856	1.01	0.40	2.50	0.990
Frequency of listening to Radio																
Not at all									1.00				1.00			
Less than once a week									0.90	0.82	0.99	0.037	0.86	0.78	0.95	0.002
At least once a week									0.92	0.83	1.01	0.089	0.92	0.83	1.02	0.099
Almost every day									1.47	1.11	1.95	0.007	1.45	1.07	1.95	0.016
Frequency of watching Television																
Not at all									1.00				1.00			
Less than once a week									1.03	0.90	1.18	0.660	1.00	0.86	1.17	0.967
At least once a week									0.95	0.84	1.06	0.333	0.90	0.80	1.02	0.104
Almost every day									1.17	0.85	1.60	0.335	1.14	0.82	1.60	0.438
*Healthcare utilization factors*																
Place of delivery																
Home													1.00			
Health facility													1.38	1.21	1.57	<0.001
Mode of delivery																
Non-caesarean													1.00			
Caesarean section ^ⱡ^													0.28	0.22	0.36	<0.001
Type of delivery assistance																
Health professional													1.00			
Traditional birth attendant.													0.77	0.60	0.98	0.037
Other untrained													0.96	0.79	1.16	0.663
No one													0.76	0.62	0.92	0.005
Antenatal Clinic visits																
None													1.00			
1–3													1.17	1.00	1.37	0.052
4+													1.20	1.01	1.41	0.034

^ divorce/separated/widowed.; ^ⱡ^ Caesarean section is a combination of both general and regional anesthesia.

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
