# Peer review of "Factors Associated with the Early Initiation of Breastfeeding in Economic Community of West African States (ECOWAS)"

_nutrients, 2019, doi:10.3390/nu11112765_

Round 1

Reviewer 1 Report

This is an interesting and important study on factors associated with early initiation of breastfeeding in Economic Community of West African States. 

Some suggestions to consider.

I appreciate you are limited to the data collected from the Demographic and Health Survey. However, is there any data on infant formula advertising spending in the countries of West Africa? Are there any midwifery practices or cultural values that interfere with EIBF that might help the readers?

Do you have any data on the  proportion of CS that are carried out under general or regional anaesthesia? I believe that EIBF is more likely post-caesarian section done under regional anaesthesia compared to general anaesthesia.

You mention the potential role of men in breast-feeding support in the conclusion as follows: "At the individual level, interventions to promote the involvement of men during pregnancy and child birth is needed in ECOWAS" .... However, male involvement in BF support is a complex and controversial topic and should be discussed earlier in the manuscript or not at all.

Yourkavitch, et al.Engaging men to promote and support exclusive breastfeeding: a descriptive review of 28 projects in 20 low- and middle-income countries from 2003 to 2013. Journal of Health, Population and Nutrition volume 36, Article number: 43 (2017

Suggested Minor Edits:

Capitalise "Factors" in the title.

 insert space ‘to 2018’ in line 5 of Abstract

Use "elective" (not elected) caesarian section in Para 5 of Discussion

It is unnecessary to qualify "crucial" with ‘very’ in the Discussion. Similarly "pain" does not need "excruciating" as a qualifier.

In Strengths and Limitations para, you say....up to 5 years of childbirth.. I presume you mean "up to 5 years after childbirth"? 

In Conclusions: "...that health promotion campaign both at the community and individual level should be effectively conducted." Consider "..that health promotion campaigns should be conducted effectively." 

Author Response

Reviewer 1

Comment

Response

1). is there any data on infant formula advertising spending in the countries of West Africa?

To our knowledge, DHS lacks information concerning infant formula advertisements via electronic or print media. However, there are dodgy television advertisements in some of the ECOWAS countries, for example, Champeny et al recently suggested that advertisements for breast milk substitutes aired frequently on television channels Dakar (Senegal) from September 2013 to September 2014 [1]. Similarly, a survey conducted in Lagos, Nigeria among mothers indicated that approximately 68% recalled observing a television advertisement for a product under the scope of the International Code of Marketing of Breast‐milk Substitutes [2].

As a result, we have inserted the following text in the limitation section

“More importantly, detailed information concerning infant formula advertising via electronic or print media was not available at the time of the DHS survey”.

1). Champeny et al. Prevalence, duration, and content of television advertisements for breast milk substitutes and commercially produced complementary foods in Phnom Penh, Cambodia and Dakar, Senegal.

2). Sanghvi, Seidel, Baker, & Jimerson, 2017. Using behavior change approaches to improve complementary feeding practices. Maternal & Child Nutrition13, e12406

2). Are there any midwifery practices or cultural values that interfere with EIBF that might help the readers?

Cultural factors that may influence mix-feed of infants were not gathered during DHS surveys in ECOWAS, hence we included cultural factors as part of our limitations earlier.

Actually, it a known practice, particularly in low and middle income countries including ECOWAS that mothers are culturally encouraged to mix-feed their infants, however the extent of the feeding practices remains unclear.

Cultural factors reported in Ghana [1] and Nigeria [2] to delay EIBF included colostrum was dirty or harmful, and perceived lack of breast milk. It was also reported in Cameroon by Kakute et al, that village elders and families pressure nursing mothers to introduce foods other than breast milk as traditional practice demands. Additionally, belief that breast milk does not increase the infants weight, belief that all family members should benefit from food grown in the family farm, and the taboo of prohibiting sexual contact during breastfeeding [3]

Refs

1). Tawiah-Agyemang C, Kirkwood BR, Edmond K, Bazzano A, Hill Z. Early initiation of breast-feeding in Ghana: barriers and facilitators. J Perinatol: official journal of the California Perinatal Association. 2008;28(Suppl 2):S46 – 52

2). Oche MO, Umar AS, Ahmed H. Knowledge and practice of exclusive breastfeeding in Kware. Nigeria Afr Health Sci. 2011;11(3):518 –23. 48.

3). Kakute et al 2005. Cultural Barriers to Exclusive Breastfeeding by Mothers in a Rural Area of Cameroon, Africa. Journal of midwifery & women's health 50(4):324-8

3).Do you have any data on the  proportion of CS that are carried out under general or regional anaesthesia?

Record regarding caesarean section in DHS data file was not categorised as general or regional anaesthesia.

However, we have added the following text as a footnote on tables 1 & 2;

“Caesarean section is a combination of both general and regional anaesthesia”.

4). I believe that EIBF is more likely post-caesarian section done under regional anaesthesia compared to general anaesthesia

Thanks for the useful information, but we are unable to measure the impact of general or regional anaesthesia in the current study because both were combined in DHS data.

5). You mention the potential role of men in breast-feeding support in the conclusion as follows: "At the individual level, interventions to promote the involvement of men during pregnancy and child birth is needed in ECOWAS" .... However, male involvement in BF support is a complex and controversial topic and should be discussed earlier in the manuscript or not at all.

We have revised this comment in the manuscript to reflect the impact of family involvement in improving EIBF especially among families from low socioeconomic group

https://familyincluded.com/breastfeeding-family-teamwork/

6). Capitalise "Factors" in the title

Thanks, we have replaced “factor” with “Factor”

 7). insert space ‘to 2018’ in line 5 of Abstract

“to2018” now reads “to 2018”

8).Use "elective" (not elected) caesarian section in Para 5 of Discussion

Thanks, we have now replaced “elected” with “elective”

9). It is unnecessary to qualify "crucial" with ‘very’ in the Discussion. Similarly "pain" does not need "excruciating" as a qualifier.

Both “very” and "excruciating" now deleted from the conclusion & discussion sections.

10). In Strengths and Limitations para, you say....up to 5 years of childbirth.. I presume you mean "up to 5 years after childbirth"? 

“...up to 5 years of childbirth” now replaced with “...up to 5 years after childbirth”

11). In Conclusions: "...that health promotion campaign both at the community and individual level should be effectively conducted." Consider "..that health promotion campaigns should be conducted effectively." 

Thanks, it now reads

"...that health promotion campaign both at the community and individual level should be considered."

Reviewer 2 Report

Overall it is a strong paper about a very important public health topic from populations that are under-represented in the literature.  My biggest concern is the validity of the cross-sectional data for EIBF (very retrospective  - remembering the exact time of first breastfeeding 2 years later!)  Is there anything in the literature to validate the accuracy of these maternal memories?

The analysis is easy to understand, but there is, in my opinion, too much data in the tables - I would put some into supplemental tables so the reader can see the important info and not get lost in the numbers...

Other more specific comments below:

Abstract: easy to understand results.  There are a few grammatical English errors in verb tenses (using the past tense when should be present) that should be corrected.

Intro: First paragraph: A few grammatical errors in using the past tense when not required.  The word “delayed” is used many times in the paper when it should be “delay” – (example – should be mothers were more likely to delay initiation of breastfeeding)

For readers not from the African continent, would be helpful to list the ECOWAS countries here.  You could then, for example, in the 3rd paragraph, remove “ECOWAS member” from the description of Nigeria.  You discuss the members later on, but until that paragraph, I was wondering exactly which countries were included, so it should be moved up to near the beginning...  Also, why 13 of 15 countries and not all?

second to last sentence: these benefits for early breastfeeding are not nutritional- should read non-nutritional, as antibodies and other benefits that come with the skin to skin contact of breastfeeding like temperature regulation as well as improved milk supply (which aren’t mentioned, but I would advise including at least the temperature regulation aspect, perhaps when you mention bonding).  The nutritional aspect would be reduced risk of hypoglycemia in at-risk infants (such as small for gestational age, which you report had lower rates of early breastfeeding but would have the most to benefit), which causes significant morbidity – I would include this aspect as well.

Third paragraph – Be careful with generalizations like “inattention to EIBF contributed to half of all under-5 deaths” – I haven’t read the entire UNICEF report but I assume they don’t specifically say this.  Could be less dogmatic, such as “could potentially contribute to half of all under-5 deaths, such as those due to prematurity or malnutrition”

Materials and Methods

2.1: Since EIBF within 1 hour is the core of your paper, would be helpful to know how/by whom this is recorded – is this retrospective? Prospective? How accurate?  [I note that you briefly discuss this in limitations, but I wanted info earlier.] And “Perceived size” of baby – is this verified by actual birth weight or objective data in any way? Or just maternal perception? Any data on how well perception correlates to objective measures?

Spelling error in “antennal” aka “antenatal”

What is “hv271” variable?  Cite a reference... 

2.2: cite a reference for Taylor series linearization method

Results

Figure 1: I assume the numbers after each country refer to the year in which the DHS was published, but this is not stated.  Also be consistent using 2 or 4 numbers for the year

Paragraph 2 (referring to table 1): In the second sentence, when you compare Nigeria and Benin, it isn’t obvious that you are simply giving an example from the table in order to explain your findings – I thought you were pulling out an important finding and was confused as to why that combination was chosen.  State, for example, when compared to children from Benin, children in Nigeria had rates of EIBG that were reduced by 58%.  I’d remove the (7.7%) and (21%) as it’s confusing and doesn’t add to the argument. 

A big question for table 1 is why everyone was compared to Benin? It doesn’t have the highest or lowest EIBF rates, it’s a rather small country – it isn’t obvious why this was chosen for your analysis as the “gold standard?”

Table 1 is hard to read and it’s not obvious what % vs %* is, as the “*” isn’t defined anywhere that I can find.  Since the table goes on for a few pages, you lose the header on top – this needs to be repeated on each page so you can follow the columns

Why is age of the child included?  Is this referring to the fact that it’s retrospective data collection (I’m guessing) so that you’d have more reporting bias for the older children?  It’s confusing in the paper to keep referring to children 0-23 months when you’re only talking about the first hour of life... [I note that you do discuss this in the limitations, but I wanted more info earlier as it kept coming up in my mind...]

Given the size of the table, I’d delete the “combined birth rank/interval” section as it’s very specific and doesn’t add a lot but takes up many lines.

Same question as above for size of baby – how is this defined? Only maternal perception?

For SE factors, why are household wealth index categories not evenly divided?  It wasn’t clear in the methods how this was calculated and this makes it more confusing.

To decrease length of the table and number of variables, would have combined some of the media categories since “almost every day” is negligible in all the categories

Table 2 – like table 1, a lot of data so hard to get the important pieces out of it.  Would very strongly consider removing or condensing some of the non-significant data (like all the birth order/rank combinations and the age of the child and some of the media info) and putting it in supplemental files

Discussion

Paragraph 1: the sentence that includes the key results (“the study indicated that place of delivery...)   - using “wealth” and saying poorest and middle is confusing and doesn’t appear to exactly mirror the results.  It appeared that the poorest mothers were less likely to practice EIBF (neg association) whereas the other factors you list are positive associations – wouldn’t put these together, or just say “wealth” and remove “poorest and middle” – this would be more accurate in that you’re saying that the not being the poorest made you more likely to practice EIBF.  Also, use “sex” not “gender” – gender is a social construct, whereas sex is a demographic description at birth.  Same argument with media exposure, etc sentence – put together the positive and negative associations in separate sentences to make it more obvious.

Third paragraph – please remove the subjective “excruciating” and would also remove the non-medical “weakness associated with difficult labour”

Fourth paragraph – the first sentence suggests the women pose a barrier – needs to be reworded.  It’s the lack of access to ANC that poses a barrier.  “However” in the next sentence should also be removed as this is in keeping with previous studies.     Same concern with “gender” – use “sex”.  And in last sentence, you havent’ defined “prelacteal” and many readers may not know what this is; also sentence needs to be revised from a grammar standpoint (the “as strong and healthy” is hanging on the end...)

5th paragraph – your results suggested that “poorer” and “middle” were more likely to practice EIBF than poorest.  Using “poor” here is confusing.  Perhaps you want to then flip your analysis and use wealthiest as the standard?

Limitations – is there any data (I mentioned this earlier) as to the accuracy of these very retrospective reports of EIBF?

Conclusions – Change gender to sex.  And again, the use of “poor” is confusing. Perhaps middle income is more accurate?  Also, the last sentence seems to come out of nowhere as far as involving men...  There’s no discussion anywhere in the paper about this so it’s an odd way to end an otherwise strong paper, with a conclusion unrelated to the results.

Author Response

Reviewer 2

COMMENT

RESPONSE

1). My biggest concern is the validity of the cross-sectional data for EIBF (very retrospective  - remembering the exact time of first breastfeeding 2 years later!)  Is there anything in the literature to validate the accuracy of these maternal memories?

The study conducted by Li et al (2015) concluded that that maternal recall is a valid and reliable estimate of breastfeeding initiation and duration, especially when the duration of breastfeeding is recalled after a short period ( 3 years).

Li, R, Scanlon, KS & Serdula, MK (2005) The validity and reliability of maternal recall of breastfeeding practice. Nutr Rev 63, 103–110.

2).The analysis is easy to understand, but there is, in my opinion, too much data in the tables - I would put some into supplemental tables so the reader can see the important info and not get lost in the numbers...

We think the presentation of tables are appropriate for an analysis that involve 13 countries and especially for readers who may be interested in doing a meta-analysis. However, we would like to defer the final decision on this suggestion to the Editor.

3). Abstract: easy to understand results.  There are a few grammatical English errors in verb tenses (using the past tense when should be present) that should be corrected.

Our understanding is that abstract should be past tense because the analysis has already happened. We also note that the revised manuscript has been extensively edited for wording and syntax.

Introduction

4).Intro: First paragraph: A few grammatical errors in using the past tense when not required.  The word “delayed” is used many times in the paper when it should be “delay” – (example – should be mothers were more likely to delay initiation of breastfeeding)

Thanks and we have revised

5). For readers not from the African continent, would be helpful to list the ECOWAS countries here.  You could then, for example, in the 3rd paragraph, remove “ECOWAS member” from the description of Nigeria. 

Agreed, we have removed the text “ECOWAS member” and included the text below at the end of the first paragraph.

ECOWAS is a regional political and economic union of fifteen countries in West Africa, founded in 1975 with members including Benin, Burkina Faso, Cabo Verde, Cote d’Ivoire, The Gambia, Ghana, Guinea, Guinea-Bissau, Liberia, Mali, Niger, Nigeria, Senegal, Sierra Leone, and Togo. The main aim of this alliance is to promote socioeconomic integration among member states to raise living standards and promote economic development, with wider implications for judicial and health service cooperation

6). You discuss the members later on, but until that paragraph, I was wondering exactly which countries were included, so it should be moved up to near the beginning...  Also, why 13 of 15 countries and not all?

Agreed and we have edited the text.

7). second to last sentence: these benefits for early breastfeeding are not nutritional- should read non-nutritional, as antibodies and other benefits that come with the skin to skin contact of breastfeeding like temperature regulation as well as improved milk supply (which aren’t mentioned, but I would advise including at least the temperature regulation aspect, perhaps when you mention bonding).  The nutritional aspect would be reduced risk of hypoglycemia in at-risk infants (such as small for gestational age, which you report had lower rates of early breastfeeding but would have the most to benefit), which causes significant morbidity – I would include this aspect as well.

Agreed and we have added the text below to the manuscript:

the nutritional benefit of EIBF may include a reduced risk of hypoglycaemia in at-risk infants (eg, small for gestational age and macrosomia infants), which can cause significant morbidity

8). Third paragraph – Be careful with generalizations like “inattention to EIBF contributed to half of all under-5 deaths” – I haven’t read the entire UNICEF report but I assume they don’t specifically say this.  Could be less dogmatic, such as “could potentially contribute to half of all under-5 deaths, such as those due to prematurity or malnutrition”

Thanks, we have replaced it with the text below

“The low rate of EIBF practice in ECOWAS could have potentially contributed to nearly half of all under-five deaths, such as those due to prematurity or malnutrition in sub-Saharan Africa region in the year 2017[2]”

Materials and methods

9). 2.1: Since EIBF within 1 hour is the core of your paper, would be helpful to know how/by whom this is recorded – is this retrospective? Prospective? How accurate?  [I note that you briefly discuss this in limitations, but I wanted info earlier.]

Agreed and we have added the text below:

“The outcome variable was EIBF. Women were asked how long after birth the baby was put to the breast for the first time. Responses were recorded in minutes and/or hours”

10). And “Perceived size” of baby – is this verified by actual birth weight or objective data in any way? Or just maternal perception? Any data on how well perception correlates to objective measures?

We did not include actual birth weight of newborns in the study analysis because almost half of the newborns were not weighed at the time of birth. As a result we used perceived newborn size at birth by mothers instead of actual birth weight because a previous study indicated that there is a close relationship between mean birth weight and perceived newborn size by the mother [21].

Ref

Titaley CR et al.: Iron and folic acid

supplements and reduced early neonatal deaths in Indonesia. Bull World Health Organ 2010, 88:500–508.

11).Spelling error in “antennal” aka “antenatal”

Corrected

12). What is “hv271” variable?  Cite a reference

Agreed and we have added the text and reference below to the manuscript.

“The hv271 is a household's wealth index value generated by the product of standardized scores (z-scores) and factor coefficient scores (factor loadings) of wealth indicators”

S. O. Rutstein, K. Johnson, “The DHS Wealth Index. DHS Comparative Reports No. 6.” (ORC Macro, Calverton, MD, 2004)

13). 2.2: cite a reference for Taylor series linearization method

Reference below was added

Rutstein SO, Rojas G. Guide to DHS statistics. Calverton, MD: ORC Macro. 2006 Sep;38.

Results

14).Figure 1: I assume the numbers after each country refer to the year in which the DHS was published, but this is not stated.  Also be consistent using 2 or 4 numbers for the year

Our apologies because this question is not clear. However, we calculated the prevalence and 95%CI reported in figure 1 – the prevalence are similar to those reported in the DHS report bur DHS don’t report CI’s

15). Paragraph 2 (referring to table 1): In the second sentence, when you compare Nigeria and Benin, it isn’t obvious that you are simply giving an example from the table in order to explain your findings – I thought you were pulling out an important finding and was confused as to why that combination was chosen.  State, for example, when compared to children from Benin, children in Nigeria had rates of EIBG that were reduced by 58%.  I’d remove the (7.7%) and (21%) as it’s confusing and doesn’t add to the argument.

We were comparing proportions between two countries and text in the manuscript now reads:

The odds of EIBF in Nigeria reduced significantly by 58% (OR=0.42, 95%CI:0.36, 0.48) compared with Benin

16). A big question for table 1 is why everyone was compared to Benin? It doesn’t have the highest or lowest EIBF rates, it’s a rather small country – it isn’t obvious why this was chosen for your analysis as the “gold standard?”

We considered Benin as the referenced category because it was the first country on the list of ECOWAS countries. Statistically, the selection of a reference category does not affect the interpretation of the results, or the reference category could be based on the research question or previous studies.

For the combined dataset, sampling weight was denormalised, and a new population-level weight was created by dividing the sampling weights by the denormalised weight. We then created a unique country-specific cluster and strata because each country had individual clusters and strata in the DHS. This was done to account for the uneven country-specific population across the organisation and to avoid the effect of countries with large population

17).Table 1 is hard to read and it’s not obvious what % vs %* is, as the “*” isn’t defined anywhere that I can find.  Since the table goes on for a few pages, you lose the header on top – this needs to be repeated on each page so you can follow the columns

Agreed and I have added a footnote.

18).Why is age of the child included?  Is this referring to the fact that it’s retrospective data collection (I’m guessing) so that you’d have more reporting bias for the older children?  It’s confusing in the paper to keep referring to children 0-23 months when you’re only talking about the first hour of life... [I note that you do discuss this in the limitations, but I wanted more info earlier as it kept coming up in my mind...]

To reduce recall bias, we restricted the analysis to the last born child in the last 2 years. For example (which may not be possible), if a woman had two children within two years, we will consider the last born child because we know mothers are less likely to forget what happens to her when she had the last child.

19). Given the size of the table, I’d delete the “combined birth rank/interval” section as it’s very specific and doesn’t add a lot but takes up many lines.

Same question as above for size of baby – how is this defined? Only maternal perception?

We think this variable is important because there is a strong correlation between birth rank and birth interval. For example, a woman had ONLY one child would report no previous birth interval and several studies have combined them.

On the question on perceived size of baby, we have responded to earlier including providing relevant references to back up our claim

20). For SE factors, why are household wealth index categories not evenly divided?  It wasn’t clear in the methods how this was calculated and this makes it more confusing.

Agreed and for clarity, I have added the text below to the manuscript.

“In the household wealth index categories, the bottom 20% of households was arbitrarily referred to as poorest households, and the top 20% as richest households”

21).To decrease length of the table and number of variables, would have combined some of the media categories since “almost every day” is negligible in all the categories

We believe that the presentation of the results is easy to follow by interested readers. Nevertheless, we would like to defer the final decision on the comment to the Academic Editor.

22).Table 2 – like table 1, a lot of data so hard to get the important pieces out of it.  Would very strongly consider removing or condensing some of the non-significant data (like all the birth order/rank combinations and the age of the child and some of the media info) and putting it in supplemental files

We believe these variables are important. For example, the role of media in breastfeeding intervention is very important. However, we would like to defer the final decision on the comment to the Academic Editor.

Discussion

23). Paragraph 1: the sentence that includes the key results (“the study indicated that place of delivery...)   - using “wealth” and saying poorest and middle is confusing and doesn’t appear to exactly mirror the results.  It appeared that the poorest mothers were less likely to practice EIBF (neg association) whereas the other factors you list are positive associations – wouldn’t put these together, or just say “wealth” and remove “poorest and middle” – this would be more accurate in that you’re saying that the not being the poorest made you more likely to practice EIBF

Thanks,

in the DHS datasets household wealth index was grouped into 5 quintiles (poorest, poorer, middle, richer and richest). As a result we adhered to the classification in our study analysis for consistency.

For clarity, we have replaced it with the text below

“The study indicated that place of delivery (health institution), mode of delivery (vaginal delivery), ANC visits (≥ 4) and household wealth index (poorer or middle-class) were significantly associated with EIBF within one hour of birth”

24). Also, use “sex” not “gender” – gender is a social construct, whereas sex is a demographic description at birth. 

“gender” replaced with “sex” in the revised manuscript

25). Same argument with media exposure, etc sentence – put together the positive and negative associations in separate sentences to make it more obvious.

For clarity, the positive association related to media exposure is now correctly stated

“Media exposure (newspaper/magazine or radio)”

26). Third paragraph – please remove the subjective “excruciating” and would also remove the non-medical “weakness associated with difficult labour”

"excruciating" now deleted from the discussion section. We also replaced “weakness associated with difficult labour” with “prolonged labour” in the revised manuscript

27). Fourth paragraph – the first sentence suggests the women pose a barrier – needs to be reworded.  It’s the lack of access to ANC that poses a barrier

Thanks, it now reads

“Previously published studies have indicated that infrequent ANC services or lack of access during pregnancy pose a significant barrier in initiating breastfeeding within one hour of post-birth[36, 37]”.

28). “However” in the next sentence should also be removed as this is in keeping with previous studies.    

“however” deleted from the revised manuscript.

29). And in last sentence, you havent’ defined “prelacteal” and many readers may not know what this is; also sentence needs to be revised from a grammar standpoint (the “as strong and healthy” is hanging on the end...)

It now reads

“However, previous studies have suggested that the practice of prelacteal feeds (or act of given foods to newborn babies before initiating breastfeeding) in male babies is common and acceptable in African[8, 46] and Asian countries[46, 47]”

30).5th paragraph – your results suggested that “poorer” and “middle” were more likely to practice EIBF than poorest.  Using “poor” here is confusing.  Perhaps you want to then flip your analysis and use wealthiest as the standard?

Thanks, we have replaced it with the text below

“Newborn babies born to mothers from a poorer or middle-class household had significantly higher odds of initiating breastfeeding within one hour of birth compared with those from poorest household”

31). Limitations – is there any data (I mentioned this earlier) as to the accuracy of these very retrospective reports of EIBF?

We have already addressed your concern with reference.

32).Conclusions – Change gender to sex.  And again, the use of “poor” is confusing. Perhaps middle income is more accurate?  Also, the last sentence seems to come out of nowhere as far as involving men...  There’s no discussion anywhere in the paper about this so it’s an odd way to end an otherwise strong paper, with a conclusion unrelated to the results.

We have adjusted most of your concern and we think it is an appropriate way to concluded a paper by first summarises the key findings and suggested way forward through some effective future interventions.
